# PRIVATELY LEARNING FROM GRAPHS WITH APPLICATIONS IN FINE-TUNING LARGE PRETRAINED MODELS

## ABSTRACT

Graphs offer unique insights into relationships and interactions between entities, complementing data modalities like text, images, and videos. By incorporating relational information from graph data, AI models can extend their capabilities beyond traditional tasks. However, relational data in sensitive domains such as finance and healthcare often contain private information, making privacy preservation crucial. Existing privacy-preserving methods, such as DP-SGD, which rely on gradient decoupling assumptions, are not well-suited for relational learning due to the inherent dependencies between coupled training samples. To address this challenge, we propose a privacy-preserving relational learning pipeline that decouples dependencies in sampled relations during training, ensuring differential privacy through a tailored application of DP-SGD. We apply this method to fine-tune large language models (LLMs) on sensitive graph data, and tackle the associated computational complexities. Our approach is evaluated on LLMs of varying sizes (e.g., BERT, Llama2) using real-world relational data from four text-attributed graphs. The results demonstrate significant improvements in relational learning tasks, all while maintaining robust privacy guarantees during training. Additionally, we explore the trade-offs between privacy, utility, and computational efficiency, offering insights into the practical deployment of our approach.

## 1 INTRODUCTION

Graph data, commonly used to represent relationships between entities, are widely employed to model complex systems in the real world (Leskovec et al., 2007; Kwak et al., 2010; Shamsi et al., 2022; Madani et al., 2022). In AI applications, the relationships captured by graph structures provide complementary information to foundation models pretrained on other modalities, such as text and images, enabling these models to more effectively handle tasks involving multiple entities (Brown et al., 2020; Dosovitskiy et al., 2020; Zhang et al., 2024; Madan et al., 2024). For instance, models trained on product descriptions or pictures may not fully capture the relationships revealed by user behaviors, such as co-purchases or co-viewings. Incorporating such relational information allows AI models to better meet users' needs, e.g., in product recommendations. Models pretrained on text or images and subsequently fine-tuned with relational information from graphs have recently found applications in various domains (Ling et al., 2023), including healthcare (Wu et al., 2021; Zhang et al., 2022; Gao et al., 2023), finance (Ouyang et al., 2024), and computer vision (Li et al., 2023a). However, the relationships involved in these applications often contain sensitive personal information, such as social connections for recommendations (Zheng et al., 2022), patient-hospital visits for clinical diagnosis (Lu & Uddin, 2023), and financial transactions for fraud detection (Kurshan & Shen, 2020). This raises critical concerns about how to protect the privacy of relational data when exposed to AI models, motivating the research in this work.

Differential Privacy (DP) (Dwork, 2006; Dwork et al., 2014) is widely considered the gold standard for measuring the privacy guarantees of data-processing algorithms (Xu et al., 2021; Pan et al., 2024). Current DP methods for model training, such as DP-SGD (Song et al., 2013; Abadi et al., 2016; Ponomareva et al., 2023), are primarily designed for tasks other than relational learning. DP-SGD, in particular, operates under the assumption that the gradient in each training step can be decoupled with respect to individual training samples that require privacy protection. Under this assumption, DP-SGD controls the norm of the gradient induced by each sample, obfuscates it by adding Gaussian noise, and thus ensures a privacy guarantee. However, relational learning on graphs

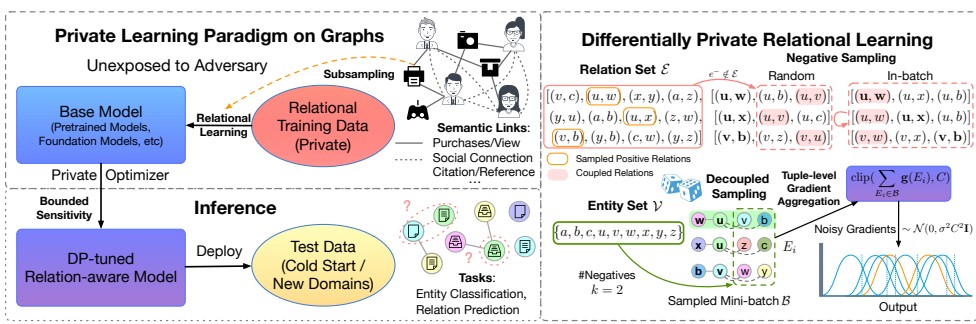

Figure 1: Learning from a domain rich in relational information (graph edges) with differential privacy and testing in new domains with limited relational information between entities. The privacy-preserving challenge in relational learning: Each loss term typically involves coupled relations through negative sampling in a mini-batch $\mathcal{B}$, where perturbing one relation (e.g., adding $(u, v)$ to or removing $(u, w)$ from the set $\mathcal{E}$) may affect multiple loss terms in the same batch. Decoupled sampling limits such perturbation to affect at most one relation tuple $E_i$ in a mini-batch.

introduces unique challenges because each loss term typically involves multiple relationships (e.g., positive and negative relationships), and each relationship involves multiple entities. Consequently, the gradient in relational learning cannot be decomposed into specific privacy-preserved samples, which violates the per-sample decoupling assumption, rendering DP-SGD not directly applicable.

Recent studies on privacy-preserving training of graph neural networks (GNNs) (Daigavane et al., 2021; Olatunji et al., 2021; Sajadmanesh & Gatica-Perez, 2021; Mueller et al., 2022; Sajadmanesh et al., 2023; Sajadmanesh & Gatica-Perez, 2024; Chien et al., 2024) do not address the issue at hand, though they also work with relational data. These works focus on training based on node classification labels, where the loss term can still be decomposed for specific nodes given the representations of these nodes output by GNNs. Their methods, which obfuscate the message-passing process to prevent privacy leakage during GNN encoding, do not mitigate privacy risks arising from relational learning—on the supervision side—where the loss term cannot be decomposed.

This study aims to introduce a privacy-preserving relational learning pipeline to address this gap. In relational learning, each loss term typically involves an observed relation (represented by edges in the graph), paired with one or more missing relations for contrast. The traditional coupled sampling of observed and missing relations means that removing or adding an observed relation will impact the gradients of multiple loss terms within the sampling batch, leading to significant privacy leakage. Our key insight is to *decouple the sampling process* for observed and missing relations. By doing so, we ensure that removing or adding an observed relation affects at most one loss term, thereby limiting the sensitivity of data perturbation in relational learning. This approach makes it theoretically compatible with the privacy accounting of the DP-SGD framework.

As an application, we apply this approach to privacy-preserving fine-tuning of large pretrained models using graph data, choosing LLMs as a proof of concept since many relational datasets involve entities with rich textual attributes. While modern privacy libraries like Opacus (Yousefpour et al., 2021), TensorFlow Privacy (McMahan et al., 2018), and JAX Privacy (Balle et al., 2022) support per-sample gradient computation for applying DP-SGD, each loss term in relational learning involves multiple entities (denoted by $K$), and each entity with textual attributes contains multiple tokens (denoted by $M$). Naively computing per-sample gradients results in keeping $\mathcal{O}(KM)$ gradient copies in memory per loss term. Even with parameter-efficient fine-tuning (PEFT) techniques like LoRA (Hu et al., 2021), modern GPUs encounter out-of-memory issues with moderate batch sizes. However, larger batch sizes are empirically preferred to enhance privacy preservation (Li et al., 2021; Anil et al., 2021; Räisä et al., 2024). To address this challenge, we propose to hook low-rank representations of individual token gradients and directly compute per-loss-term gradients, thereby eliminating the instantiation of $\mathcal{O}(KM)$ gradients. This approach significantly alleviates memory constraints, enabling more efficient privacy-preserving fine-tuning.

We evaluate our approach by testing whether LLMs can learn from private domains rich in relational data and enhance performance on relational learning tasks in new domains that lack relational information between entities, which often occurs in cold-start recommendation (Bobadilla et al.,

2012) and zero-shot relational learning (Cai et al., 2024). Using real-world relational data from four text-attributed graphs, we fine-tune BERT (Devlin et al., 2018) and Llama2 (Touvron et al., 2023) at various model sizes (110M, 340M, 7B) under different levels of DP ($\epsilon \leq 10$) to stimulate two use cases of cross-category co-purchase recommendation and cross-regional model deployment. Our results demonstrate that LLMs can effectively learn from relational data to address relational learning tasks, even when working with sensitive data that requires DP guarantees. Additionally, we investigate the trade-offs between utility, privacy, and computational efficiency in LLM-based relational learning, extending existing research of privacy-preserving learning with LLMs on standard (non-relational) text data (Li et al., 2021). These findings offer valuable insights for the practical deployment of LLMs in privacy-preserving relational learning scenarios.

## 2 PRELIMINARIES: NOTATIONS AND STANDARD LEARNING VIA DP-SGD

Graph $(\mathcal{V}, \mathcal{E}, X)$ consists of a relation set $\mathcal{E}$ that describes the relationship between entities in $\mathcal{V} = [N]$. Each entity $v \in \mathcal{V}$ is associated with an attribute $X_v$ of text, images, or other data modalities.

**Definition 2.1** (($\epsilon, \delta$)-Differential Privacy). A randomized mechanism $\mathcal{M}$ satisfies an ($\epsilon, \delta$)-differential privacy if for any adjacent datasets $\mathcal{D}, \mathcal{D}'$ that differ in one sample, and any output set $S \subset \text{Range}(\mathcal{M})$, $\Pr(\mathcal{M}(\mathcal{D}) \in S) \leq \exp(\epsilon) \Pr(\mathcal{M}(\mathcal{D}') \in S) + \delta$, where $\varepsilon, \delta \geq 0$ measure the privacy loss. Smaller values of $\epsilon, \delta$ imply stronger privacy guarantees.

The notion of adjacent datasets can be generalized to relational data. Specifically, two relation sets $\mathcal{E}, \mathcal{E}'$ are considered adjacent if one can be obtained from the other by adding or removing a relation. We provide the formal definition of DP guaranteed in this work in Sec. 3.1.

**Standard DP Learning Paradigm.** To achieve data privacy for training deep learning models, DP-SGD (see Alg. 2, Song et al. (2013); Abadi et al. (2016)) was proposed. Consider a mini-batch $\mathcal{B}$ with $b$ samples. The model parameter $\Theta$ is updated iteratively as $\Theta_{t+1} = \Theta_t - \eta \mathbf{g}_t(\mathcal{B})$, where $\eta$ is the learning rate, and $\mathbf{g}_t(\mathcal{B}) = \partial \ell(\Theta_t; \mathcal{B})/\partial \Theta_t$ is the gradient of the loss $\ell$ on $\mathcal{B}$ w.r.t the parameters $\Theta_t$ at step $t$. Adding or removing one sample from $\mathcal{B}$ can change $\mathbf{g}(\mathcal{B})$, causing privacy leakage that can be measured by the *sensitivity* $\Delta_2 = \max_{\mathcal{B}, \mathcal{B}'} ||\mathbf{g}(\mathcal{B}) - \mathbf{g}(\mathcal{B}')||_2$, where $\mathcal{B}'$ and $\mathcal{B}$ are different in one sample $|(\mathcal{B} \backslash \mathcal{B}') \cup (\mathcal{B}' \backslash \mathcal{B})| = 1$. DP-SGD first clips *per-sample* gradients to control the sensitivity and then adds Gaussian noise to obfuscate the potential change to achieve DP,

$$\tilde{\mathbf{g}}(\mathcal{B}) = \frac{1}{b} \left[ \sum_{x_i \in \mathcal{B}} \text{Clip}(\mathbf{g}(x_i), C) + \mathcal{N}(0, \sigma^2 C^2 \mathbf{I}) \right],$$

where $\mathbf{g}(x_i)$ is the parameter gradient of the loss on example $x_i$, $\text{Clip}(\mathbf{g}, C) = \mathbf{g}/\max(1, ||\mathbf{g}||_2/C)$ for some constant $C > 0$. Clipping per-sample gradients limits the sensitivity to at most $C$. Then, the Gaussian noise with standard deviation $\sigma C$ is added to achieve DP based on the Gaussian mechanism for this step (Dwork et al., 2014). To obtain the DP guarantee for the entire training procedure, the composition theorem (Balle & Wang, 2018) is used to account for the total privacy loss over $T$ steps. Mini-batch sampling also allows for some privacy amplification, for which interested readers may check relevant works for more details (Balle et al., 2018; Wang et al., 2019).

## 3 METHODOLOGY

In this section, we first introduce the technical difficulty of applying standard DP-SGD when training models in relational learning. Then, we propose a pipeline that addresses this difficulty and can provably achieve differential privacy in learning from the relational data. To apply our proposed pipeline to fine-tune large pretrained language models on text-attributed graphs, we further address the computing challenge induced by the control of gradient sensitivity.

### 3.1 CHALLENGES IN PRIVATE RELATIONAL LEARNING

**Enhance Models with Relational Data** Relational data provide complementary information to models trained on a specific modality, enabling them to more effectively handle tasks involving multiple entities. Suppose the representation of each entity $u$ is obtained from a model parameterized

by $\Theta$ encoding its attribute, i.e., $\mathbf{h}_u = f_\Theta(X_u)$. A common approach of relational learning is to use relationships between entities to refine their representations (Yasunaga et al., 2022b; Duan et al., 2023; Xie et al., 2023). This is typically achieved via training based on a loss $\ell$ that can be generally written as the following form (Hadsell et al., 2006; Schroff et al., 2015; Song et al., 2015; Sohn, 2016; Ying et al., 2018; Oord et al., 2018). Given a tuple $E_i$, consisting of an observed (positive) relation $e_i^+ \in \mathcal{E}$ and several missing (negative) relations $\{e^- i_j\}_{j=1}^k$ where $e_{i_j}^- \notin \mathcal{E}$, the loss is denoted as $\ell(\Theta; E_i)$. For a mini-batch $\mathcal{B}$ of tuples, the loss sum for $\mathcal{B}$ is computed as

$$\mathcal{L}_\Theta(\mathcal{B}) = \sum_{E_i \in \mathcal{B}} \ell(\Theta; E_i) = \sum_{E_i \in \mathcal{B}} \ell(\Theta; (e_i^+, \{e_{i_1}^-, \ldots, e_{i_k}^-\})). \tag{1}$$

For convenience, let $\mathbf{z}_e = \Gamma(\mathbf{h}_u, \mathbf{h}_w)$ denote the combined representations of entities in each relationship. One popular choice of $\ell$ is the InfoNCE loss (Oord et al., 2018): $\ell(\Theta; E_i) = -\ln\left(\exp(\mathbf{z}_{e_i^+})/\sum_{e' \in E_i} \exp(\mathbf{z}_{e'})\right)$. Another choice is the pairwise Hinge loss $\ell(\Theta; E_i) = [\gamma + \mathbf{z}_{e_i^+} - \mathbf{z}_{e_{i_j}^-}]_+$ which is commonly used for learning from complex multi-relations in knowledge graphs (Bordes et al., 2013; Wang et al., 2014; Yang et al., 2014; Lin et al., 2015). Here, $\gamma$ represents the margin, and $\mathbf{z}_e$ also encodes the representation of the relationship besides the entities. Note that our method for relational learning may even be extended to the case where each relationship contains more than two entities, such as network motifs (Milo et al., 2002; Benson et al., 2016) and hyperedges (Berge, 1984), although the later discussion focuses on pairwise relationships.

**Relational Learning with Different Privacy**    For relational learning, the information subjected to be protected is *the existence of a relation $e$* in the relation set $\mathcal{E}$, formally defined as follows.

**Definition 3.1.** (DP for Relational Data) An $(\epsilon, \delta)$-DP algorithm for relational data ensures that the output obtained from a randomized mechanism $\mathcal{M} : \mathcal{X} \to \mathcal{Y}$ for any adjacent relation sets $\mathcal{E}, \mathcal{E}' \sim \mathcal{X}$ and measurable sets $Y \subset \mathcal{Y}$ satisfy: $\Pr[\mathcal{M}(\mathcal{E}) \in Y] \le e^\epsilon \Pr[\mathcal{M}(\mathcal{E}') \in Y] + \delta$.

Achieving DP for relational data limits the ability of the *best possible* adversary to uncover any specific relationship between entities used for training from the model parameters. When the set of relations is defined by a plain graph, the above concept reduces to the definition of edge-level DP widely used in privacy-preserving graph algorithms (Hay et al., 2009).

Recall that DP-SGD relies on clipping per-sample gradients to bound the sensitivity of the gradient sum of a mini-batch. For relational learning, the gradient sum $\mathbf{g}(\mathcal{B})$ of mini-batch $\mathcal{B}$ is given by

$$\mathbf{g}(\mathcal{B}) = \frac{\partial \mathcal{L}_\Theta(\mathcal{B})}{\partial \Theta} = \sum_{E_i \in \mathcal{B}} \mathbf{g}(E_i) = \sum_{E_i \in \mathcal{B}} \left[ \underbrace{\frac{\partial \ell(\Theta; E_i)}{\partial \mathbf{z}_{e_i^+}} \cdot \frac{\partial \mathbf{z}_{e_i^+}}{\partial \Theta}}_{\text{Positive Relation}} + \underbrace{\sum_{j=1}^k \left( \frac{\partial \ell(\Theta; E_i)}{\partial \mathbf{z}_{e_{i_j}^-}} \cdot \frac{\partial \mathbf{z}_{e_{i_j}^-}}{\partial \Theta} \right)}_{\text{Negative Relations}} \right]. \tag{2}$$

The challenge comes from the fact that practical sampling of negative relations is usually coupled with positive relations in the same mini-batch. As a result, removing or adding a positive relation $e \in \mathcal{E}$ will not only change the tuple $E_i$ that contains $e$ but also potentially affect other tuples in $\mathcal{B}$. The impact on *multiple* terms in the sum of gradients in Eq. (2) prohibits us from properly controlling the sensitivity of $\mathbf{g}(\mathcal{B})$ by clipping each individual gradient $\mathbf{g}(E_i)$.

Specifically, for a mini-batch $\mathcal{B}$, negative relations in Eq. (1) are typically sampled by two methods (also illustrated in Fig. 1, Right): *Random Negative Sampling* is a widely used method for negative sampling (Yang et al., 2024). Given a positive relation $e_i^+ = (u, w)$, it uniformly samples negative relations containing either entity $u$ or $w$ from the complement set $\bar{\mathcal{E}} = \binom{\mathcal{V}}{2} \backslash \mathcal{E}$, e.g., $e_{i_j}^- = (u, v) \in \bar{\mathcal{E}}$. This method requires access to $\mathcal{E}$ to compute $\bar{\mathcal{E}}$ for negative sampling and makes the sampled negative relations dependent on the positive relations that share common entities. If an originally negative relation $(u, v)$ is added as a positive relation to $\mathcal{E}$, all tuples in $\mathcal{B}$ that previously sampled $(u, v)$ as negative relations will change. In the worst case, it may affect the entire mini-batch, introducing large sensitivity that cannot be properly controlled via per-sample gradient clipping. *In-batch Negative Sampling* is another even more widely adopted method for training large models due to its computational efficiency (Chen et al., 2020; You et al., 2020; Gao et al., 2021) but suffers from a similar issue. It does not need the access of $\mathcal{E}$ for negative sampling. Instead, it implicitly samples

negatives by pairing all other positive relations sampled in the same mini-batch with one end of $e_i^+$ as negative relations. This could be as bad as impacting the whole mini-batch when perturbing one positive relation: If a positive relation $e_i^+ \in \mathcal{B}$ is removed, the loss of every other tuple $E_j$ in $\mathcal{B}$ will be impacted as the entities in $e_i^+$ may be used to form the negative relations in $E_j$.

## 3.2 PRIVACY-PRESERVING RELATIONAL LEARNING

To address the challenges, we propose to decouple the negative sampling from the set of positive relations. The idea is simple but effective. Specifically, for each tuple $E_i$, we sample negative relations for contrast by randomly pairing one end of the positive relation $e_i^+$ with entities sampled uniformly at random from the whole entity set $\mathcal{V}$. This method neither needs access to the relation set $\mathcal{E}$ nor leverages other positive relations in the same mini-batch for negative sampling, which eliminates the coupling effect aforementioned in Sec. 3.1. Note that this pairing strategy may generate negative relations $(u, v)$ that are actually positive relations $(u, v) \in \mathcal{E}$ but with a low probability. Fortunately, our experiments show this does not obviously hurt the model performance.

Now, removing or adding a positive relation will change *at most one* tuple $E_i$ in a mini-batch, and hence, by clipping the norm of the gradient of each tuple $\mathbf{g}(E_i)$, we are able to bound the sensitivity of the gradient sum $\mathbf{g}(\mathcal{B})$: The $k$-many negative relations $\{e_{i_j}^-\}_{j=1}^k$ also contribute to the gradient computation $\mathbf{g}(\mathcal{B})$, but in this new strategy, they only depend on the positive relation $e_i^+$ in the same tuple and their effect is bounded through clipping $\mathbf{g}(E_i)$. This sampling method is compatible with DP-SGD: Each aggregated gradient $\mathbf{g}(E_i)$ in a mini-batch is clipped and noised as

$$\tilde{\mathbf{g}}(\mathcal{B}) = \frac{1}{b} \left[ \sum_{E_i \in \mathcal{B}} \mathrm{Clip}\left(\mathbf{g}(E_i), C\right) + \mathcal{N}(0, \sigma^2 C^2 \mathbf{I}) \right]. \tag{3}$$

With decoupled negative sampling and gradient obfuscation via Eq. (3), the privacy analysis of standard DP-SGD holds for relational learning, since each relation $e \in \mathcal{E}$ influences the gradient sum at most $C$. The full pipeline to achieve $(\epsilon, \delta)$-DP for relational learning is described in Alg. 1.

However, per-sample gradients are practically hard to compute as the gradient coming from each training sample needs to be properly tracked. This becomes even more challenging in relational learning. Modern privacy libraries such as Opacus (Yousefpour et al., 2021) support hooking the parameter gradient through a training sample when one sample takes only one data point. However, for relational learning, they can be only used to hook the parameter gradient through each entity $\mathbf{g}(u|e', E_i) = \frac{\partial \ell(\Theta; E_i)}{\partial \mathbf{z}_{e'}} \cdot \frac{\partial \mathbf{z}_{e'}}{\partial \mathbf{h}_u} \cdot \frac{\partial \mathbf{h}_u}{\partial \Theta}$ during a backward pass (Yousefpour et al., 2021). This means that the gradient $\mathbf{g}(E_i)$ of model parameters through one tuple needs to be calculated through multiple entities in this tuple, i.e., $\mathbf{g}(E_i) = \sum_{e' \in E_i} \sum_{u \in e'} \mathbf{g}(u|e', E_i)$. Computing and caching each $\mathbf{g}(u|e', E_i)$ incurs significant computational overhead for tuples of large sizes $k$. This issue becomes more serious for training large models. Next, we aim to address this computational problem.

## 3.3 EFFICIENT PRIVACY COMPUTING IN RELATIONAL LEARNING

Modern graph and relational datasets involve entities with rich textual attributes (Jin et al., 2023a), which makes finetuning LLMs a great application for our privacy-preserving relational learning pipeline. However, this introduces a further challenging issue in computation when we work with LLMs. Specifically, when applying DP-SGD to the models that take in multiple tokens, such as Transformers (Vaswani et al., 2017), the parameter gradient through each token prediction will be hooked. This means when our method is applied to LLMs, the parameter gradient through each token $m$ in each entity $u$ is actually hooked, which introduces huge memory consumption. Prior works (Lee & Kifer, 2021; Li et al., 2021) proposed some strategies to address this issue for LLM finetuning on plain text data, but we find that these techniques are insufficient for relational learning: Relational learning introduces another dimension of tuple size $k$ as discussed in Sec. 3.2.

Next, we present a customized approach for efficiently computing the per-tuple gradient $\mathbf{g}(E)$ for linear and embedding layers of Transformers in relational learning, which leverages the low-rank characterization of per-sample gradient (Goodfellow, 2015) and the structure of the per-tuple gradient $\mathbf{g}(E)$ in relational learning.

**Algorithm 1:** Model Fine-tuning on Relational Data with Differential Privacy

**Input:** pretrained model $f_\Theta$ (e.g., LLMs), graph $\mathcal{G} = (\mathcal{V}, \mathcal{E}, X)$, scoring function $\Gamma$, loss function $\ell$; **Parameters:** learning rate $\eta_t$, batch size $b$, number of negative samples $k$, gradient norm threshold $C$, noise multiplier $\sigma$ or privacy budget $\epsilon$.
**Initialize** find the optimal value of $\sigma$ via calibration if $\epsilon$ is given.
**for** $t = 1$ **to** $T$ **do**
    **Subsampling**
I. Randomly sample $\mathcal{B}_t$ from $\mathcal{E}$ with sampling ratio $b/|\mathcal{E}|$.
II. For each sampled positive relation $e_i^+$ in the batch, randomly sample $k$ entities $(v_{i_1}, \ldots, v_{i_k})$ without replacement from $\mathcal{V}$ and pair them with one end of $e_i^+$ as negatives $\{e_{i_j}^-\}_{j=1}^k$, which forms a tuple of $k+1$ relations as $E_i = (e_i^+, \{e_{i_j}^-\}_{j=1}^k)$.
    **Compute & Aggregate Gradient**
$\mathbf{g}_t(E_i) = \sum_{e' \in E_i} \sum_{u \in e'} \frac{\partial \ell(\Theta; E_i)}{\partial \mathbf{z}_{e'}} \cdot \frac{\partial \mathbf{z}_{e'}}{\partial \mathbf{h}_u} \cdot \frac{\partial \mathbf{h}_u}{\partial \Theta}$, where $\mathbf{z}_{e'} = \Gamma(\mathbf{h}_u, \mathbf{h}_v)$ for relation $e' = (u, v)$ and $\mathbf{h}_u = f_\Theta(X_u)$ for entity $u$.
    **Gradient Clipping & Add Privacy Noise & Update Parameters**
$\tilde{\mathbf{g}}_t \leftarrow \frac{1}{b} \left[ \sum_{E_i \in \mathcal{B}_t} [\mathbf{g}_t(E_i) / \max(1, ||\mathbf{g}_t(E_i)||_2 / C)] + \mathcal{N}(0, \sigma^2 C^2 \mathbf{I}) \right]$
$\Theta_{t+1} \leftarrow \Theta_t - \eta_t \tilde{\mathbf{g}}_t$
**end for**
**Output** $\Theta_T$ and calculate the overall privacy cost $(\epsilon, \delta)$ using an accounting method if $\sigma$ is given.

For a linear layer in Transformers, its weight matrix is $\mathbf{W} \in \mathbb{R}^{p \times d}$, where $d, q$ are the input and output dimensions, respectively. For a tuple $E$, let $\mathbf{a} \in \mathbb{R}^{K \times M \times d}$ denote the concatenated input, which contains $K = 2(k+1)$ entities, and each entity is associated with $M$ tokens. Let $\mathbf{s} \in \mathbb{R}^{K \times M \times p}$ be the output, where $\mathbf{s}_{i,j} = \mathbf{W}\mathbf{a}_{i,j}$ corresponds to the $j$-th token of the $i$-th entity in the tuple $E$. Denote the gradient w.r.t. $\mathbf{s}_{i,j}$ as $\mathbf{r}_{i,j} = \frac{\partial \ell(\Theta; E)}{\partial \mathbf{s}_{i,j}}$. Then, the gradient of $\mathbf{W}$ through $\mathbf{s}_{i,j}$ can be represented as $\nabla_{\mathbf{W}|\mathbf{s}_{i,j}} \ell = \frac{\partial \ell(\Theta; E)}{\partial \mathbf{s}_{i,j}} \cdot \frac{\partial \mathbf{s}_{i,j}}{\partial \mathbf{W}} = \mathbf{r}_{i,j} \mathbf{a}_{i,j}^T \in \mathbb{R}^{p \times d}$. To compute the per-tuple gradient w.r.t. $\mathbf{W}$, i.e., $\sum_{i=1}^K \sum_{j=1}^M \nabla_{\mathbf{W}|\mathbf{s}_{i,j}} \ell$, it is costly to first compute $\mathbf{r}_{i,j} \mathbf{a}_{i,j}^T$ for each token and then compute the sum. Instead, a cheaper way is to record $\mathbf{r} = [\cdots, \mathbf{r}_{i,j}, \cdots] \in \mathbb{R}^{K \times M \times p}$ and $\mathbf{a} = [\cdots, \mathbf{a}_{i,j}, \cdots] \in \mathbb{R}^{K \times M \times d}$, and compute $\mathbf{r}\mathbf{a}^\top$ to accomplish the sum. This strategy can reduce the memory cost from $\mathcal{O}(KMpd)$ to $\mathcal{O}(KM(p+d) + pd)$. In our experiments that use Llama2-7B (Touvron et al., 2023), $K \in [10, 34]$ and $M = 32$ while $p = d = 4096$ in attention blocks and $p = 32000, d = 4096$ in the embedding block. So, $pd \gg KM(p+d)$ and thus the overall saving based on the above approach is a factor of $\mathcal{O}(KM)$. In addition, some PEFT techniques such as LoRA (Hu et al., 2021) can be incorporated into the pipeline to further reduce the memory cost to $\mathcal{O}(KM(p+d+2r) + (p+d)r)$, where $r$ is the rank of adjustment $\triangle \mathbf{W}$ for parameter $\mathbf{W}$.

## 4  EXPERIMENTS

**Problem Setting & Datasets**  Our experiment design aims to simulate widely encountered scenarios in relational learning, where the relational data used for enhancing models contain sensitive or proprietary information that needs to be protected, such as in applications of e-commerce (Peng et al., 2024), finance (Wu et al., 2023; Ouyang et al., 2022), and healthcare (Gao et al., 2023). We consider two specific use cases: *Cross-category recommendation* - When launching new product lines, RecSys models often face the problem of lacking historical data for prediction (e.g., co-purchase), which can be alleviated by leveraging user purchase history of complementary categories, but these co-purchase relations contain sensitive user behaviors. *Cross-regional model deployment* - Financial institutions operate in multiple locations, and their service models (e.g., fraud detection) are normally trained on transaction data collected from major markets and then deployed to multiple regions after fine-tuning, but this practice is often challenged by regional data protection regulations.

To simulate these scenarios, we focus on feature-rich real-world datasets and select two publicly available text-attributed graphs with millions of entities/relations: the e-commerce network from Amazon (AMAZ) (McAuley et al., 2015) and the academic network from Microsoft Academic Graph (MAG) (Sinha et al., 2015). In the AMAZ dataset, each entity is a shopping item, and the

Table 1: Dataset statistics and experimental setup for evaluation.

| Dataset | #Entity | #Relation | #Entity (Test) | #Classes | #Relation (Test) | Test Domain |
|---------|---------|-----------|----------------|----------|------------------|-------------|
| AMAZ-Cloth | 960,613 | 4,626,125 | 476,510 | 9 | 10,000 | AMAZ-Sports |
| AMAZ-Sports | 357,936 | 2,024,691 | 129,669 | 16 | 10,000 | AMAZ-Cloth |
| MAG-USA | 132,558 | 702,482 | 6,653 | 40 | 63,635 | MAG-CHN |
| MAG-CHN | 101,952 | 285,991 | 6,534 | 40 | 34,603 | MAG-USA |

relation between them indicates that they are co-purchased by customers. AMAZ is divided into two domains based on the item category: clothing and sports. In the MAG dataset, each entity is a research paper, and the relation between them reflects one cites the other. MAG is split into two domains based on the region of main authors: USA and China. In total, four domain-specific subgraphs (see Table 1) are used to privately fine-tune models through relational learning and benchmark their performance on the corresponding test domains for relation prediction and entity classification.

The following questions are to be answered for private relational learning:

- **Q1** Can the target model privately fine-tuned on relations from the training graph learn generalizable relational knowledge and benefit downstream relational learning tasks on new test domains?

- **Q2** How does the parameter in relational learning - negative sampling size $k$ impact the results? How does the selection of other hyperparameters, such as the batch size, the learning rate, and privacy hyperparameters $\sigma$, $C$, impact the results in relational learning? Does it follow the same principles in the non-relational learning (Li et al., 2021)?

### 4.1 EXPERIMENTAL SETTINGS

**Pretrained Models** Off-the-shelf pretrained language models: BERT (Devlin et al., 2018), a language model pretrained with masked language modeling (MLM) and next sentence prediction objectives on Wikipedia and BookCorpus, with parameters of 110M (base) and 340M (large). SciBERT (Beltagy et al., 2019) is trained on 1.14M paper abstracts and full text from Semantic Scholar under the same pertaining strategies as BERT. LinkBERT (Yasunaga et al., 2022b) is pretrained with MLM as BERT and the relation-based objective for predicting linked documents. Note that some documents and relations in the MAG dataset may be used during the pretraining of SciBERT and LinkBERT, which potentially causes some data leakage. Llama2-7B (Touvron et al., 2023) is one of the most popular open-source pretrained and fine-tuned LLMs with 7 billion parameters.

**Task Settings** The public pretrained models are fine-tuned under the supervision of relational information by Alg. 1 with the InfoNCE loss and DP-Adam (see Alg. 3) [1]. The privacy loss is tracked through PRV accounting (Gopi et al., 2021). Following existing work on private fine-tuning of LLMs (Li et al., 2021; Yu et al., 2021), we consider privacy levels $\epsilon \in \{4, 10\}$ and $\delta = \frac{1}{|\mathcal{E}_{\text{train}}|}$ for a training set of size $|\mathcal{E}_{\text{train}}|$. We tune hyperparameters based on the InfoNCE loss under given privacy parameters. Privately fine-tuned models are deployed to the corresponding test domains (e.g., trained on relations from AMAZ-Cloth and tested on AMAZ-Sports) under the settings of zero-shot and 16-shot for relation prediction, and 8-shot for entity classification. For relation prediction, we use ranking metrics of top@1 precision (PREC@1) and mean reciprocal rank (MRR) to evaluate each model on in-batch negative samples with a batch size of 256, the same as Jin et al. (2023b). For entity classification, Macro-F1 and Micro-F1 are used. Other details are left in Appx. C.

**Baselines** To the best of our knowledge, our approach is the first for relational learning with differential privacy. To compare with relevant and feasible privacy-preserving techniques that satisfy DP for relational data, we apply the standard randomized response (RR) baseline to the relation set $\mathcal{E}$ and then perform model fine-tuning on the processed relation set that achieves $\epsilon$-DP. Given an entity $u$, for each pair $(u, v), v \in \mathcal{V}, v \neq u$, we apply the randomized response mechanism (Dwork et al., 2014): with probability $p = 1/(1 + \exp(\epsilon))$, the relation label of $(u, v)$ is flipped; otherwise, the original label is kept. Note that this baseline requires $\Theta(N^2)$ time complexity and drastically increases the number of relations for smaller $\epsilon$, which greatly limits its applicability.

---

[1] DP has the post-processing property (Dwork et al., 2014), resulting in the same privacy guarantees for DP-SGD and DP-Adam using the same obfuscated gradient information after the Gaussian mechanism. We use DP-Adam as the default optimizer as in previous works (Li et al., 2021; Yu et al., 2021).

Table 2: Results on **zero-shot** relation prediction with private relational learning.

| Privacy | Method | MAG-USA | | MAG-CHN | | AMAZ-Cloth | | AMAZ-Sports | |
| | | PREC@1 | MRR | PREC@1 | MRR | PREC@1 | MRR | PREC@1 | MRR |
|---|---|---|---|---|---|---|---|---|---|
| base model zero-shot | BERT.base | 4.41 | 9.94 | 6.48 | 12.69 | 14.90 | 22.41 | 8.36 | 14.04 |
| | BERT.large | 2.00 | 5.48 | 2.71 | 6.39 | 5.72 | 10.11 | 3.78 | 7.37 |
| | SciBERT | 8.70 | 17.12 | 13.89 | 23.96 | - | - | - | - |
| | LinkBERT.large | 1.09 | 4.01 | 1.46 | 4.75 | 4.01 | 8.60 | 2.06 | 5.37 |
| | Llama2-7B | 4.24 | 8.68 | 5.21 | 9.71 | 19.45 | 27.41 | 6.13 | 10.11 |
| $\epsilon = \infty$ | BERT.base | 28.07 | 39.11 | 41.93 | 53.91 | 36.13 | 47.07 | 29.84 | 39.61 |
| | BERT.large | 26.37 | 37.73 | 40.90 | 53.16 | 36.89 | 47.50 | 29.30 | 39.76 |
| | Llama2-7B | 32.80 | 46.67 | 45.65 | 58.59 | 41.01 | 52.39 | 29.21 | 41.44 |
| $\epsilon = 10$ (RR) | BERT.base | 3.28 | 8.70 | 5.10 | 11.47 | 19.97 | 29.76 | 8.03 | 13.73 |
| | BERT.large | 5.67 | 11.75 | 8.65 | 15.43 | 22.81 | 32.31 | 7.36 | 12.15 |
| | Llama2-7B | 13.64 | 22.33 | 9.92 | 16.67 | 30.39 | 41.48 | 19.63 | 27.66 |
| $\epsilon = 10$ (Ours) | BERT.base | 23.29 | 33.98 | 35.64 | 47.74 | 32.63 | 43.17 | 26.66 | 36.76 |
| | BERT.large | 22.71 | 33.76 | 35.18 | 47.03 | 31.20 | 41.28 | 28.18 | 38.68 |
| | Llama2-7B | 24.07 | 37.53 | 34.58 | 48.76 | 40.16 | 51.25 | 29.54 | 39.90 |
| $\epsilon = 4$ (Ours) | BERT.base | 22.08 | 32.69 | 31.42 | 43.54 | 33.24 | 43.67 | 26.82 | 36.80 |
| | BERT.large | 21.78 | 32.60 | 34.84 | 46.62 | 29.73 | 39.63 | 27.63 | 38.06 |
| | Llama2-7B | 22.55 | 35.47 | 32.50 | 46.68 | 39.67 | 51.09 | 29.25 | 39.35 |

## 4.2 EVALUATION OF PRIVATELY FINE-TUNED MODELS

In this section, we study the performance of pretrained language models privately fine-tuned on text-attributed graphs for relation prediction and entity classification on new test domains. The scale of privacy noise $\sigma$ and the exact privacy loss $\epsilon$ on relational data used for training each model are reported in Table 8, Appx. D.

*Relation Prediction* aims to estimate the likelihood of forming a relationship between two entities with specific semantics. Under the *zero-shot* setting, all pretrained language models are privately fine-tuned on relations from the training graph and then are directly deployed on the test domain for inference. This is often faced in cold-start recommendation problems, where the test domain lacks relational information. Results of zero-shot relation prediction in Table 2 show that using co-purchase/citation relations from training graphs to fine-tune language models through our approach can improve their base models' performance on new test domains under DP guarantee $\epsilon = \{4, 10\}$. There is only a modest performance drop compared to the non-private fine-tuned baselines ($\epsilon = \infty$), which is much smaller than all training on relational sets processed by the randomized response mechanism (not computationally feasible for $\epsilon = 4$). This observation validates the effectiveness of privacy-preserving relational learning. Decoder-only LLMs tend to perform worse than encoder models in embedding text (Li & Li, 2023; BehnamGhader et al., 2024), as reflected in the comparison between their base models in Table 2. Through (private) relation learning, Llama2-7B can also generate rich contextual representations to predict relations and outperform the widely used BERT-based encoder. Next, we consider the *few-shot* setting used for cases like cross-regional model deployment, which is often limited by resource or relational data scarcity. Here, the model obtained above is fine-tuned using 16 training and 16 validation relations from the test domain. Table 3 shows that if further few-shot fine-tuning is allowed, privately fine-tuned language models still outperform their base models, in particular providing better performance on the MAG dataset than SciBERT/LinkBERT models pretrained on documents and their relations in scientific domains.

*Entity Classification* This task aims to investigate whether injecting relational information helps language models classify text-attributed entities in adjacent new domains. This is motivated by the above relation prediction results, where introducing structural knowledge between entities can go beyond contextual semantics and help models refine their internal representations of entities across domains. Here, the language model is used as an encoder, and a classifier is attached to take entity embeddings as input for classification. We freeze the parameters of language models and only use few-shot examples to initialize the classifier. The entity classes are coarse-grained category names from AMAZ and MAG networks, where 8 labeled training and 8 validation entities of each class are used for training, and thousands of new entities are used for testing. Table 4 shows the quality of entity embeddings from the models privately fine-tuned on relational data is better than those directly generated from their base models, except for AMAZ-cloth. The performance drop on AMAZ-cloth is due to the potential misalignment between the objective of relation-based fine-tuning and entity

Table 3: Results on **16-shot** relation prediction with private relational learning.

| Privacy | Model | MAG-USA | | MAG-CHN | | AMAZ-Cloth | | AMAZ-Sports | |
| | | PREC@1 | MRR | PREC@1 | MRR | PREC@1 | MRR | PREC@1 | MRR |
|---|---|---|---|---|---|---|---|---|---|
| base model few-shot | BERT.base | 10.24 | 18.94 | 17.10 | 27.84 | 20.42 | 29.74 | 14.70 | 23.46 |
| | BERT.large | 6.57 | 13.88 | 9.61 | 17.75 | 19.57 | 28.69 | 11.23 | 17.80 |
| | SciBERT | 22.27 | 34.24 | 32.42 | 46.10 | - | - | - | - |
| | LinkBERT.large | 21.76 | 31.93 | 35.09 | 47.80 | 13.41 | 19.24 | 23.21 | 30.95 |
| | Llama2-7B | 6.21 | 12.26 | 6.29 | 11.51 | 20.25 | 28.42 | 7.17 | 11.79 |
| $\epsilon = \infty$ | BERT.base | 27.28 | 38.61 | 39.15 | 51.28 | 33.45 | 44.42 | 29.57 | 39.71 |
| | BERT.large | 26.19 | 37.69 | 37.91 | 49.93 | 34.60 | 45.48 | 29.85 | 40.79 |
| | Llama2-7B | 35.45 | 49.30 | 45.89 | 58.84 | 41.42 | 52.59 | 31.92 | 44.83 |
| $\epsilon = 4$ (Ours) | BERT.base | 24.56 | 35.55 | 33.62 | 45.72 | 33.40 | 44.23 | 28.64 | 38.34 |
| | BERT.large | 23.09 | 34.21 | 37.23 | 48.65 | 30.39 | 40.78 | 27.80 | 37.87 |
| | Llama2-7B | 22.88 | 35.94 | 32.07 | 46.22 | 39.94 | 51.10 | 29.78 | 40.27 |

Table 4: Results on **8-shot** entity classification with private relational learning.

| Privacy | Model | MAG-USA | | MAG-CHN | | AMAZ-Cloth | | AMAZ-Sports | |
| | | Macro-F1 | Micro-F1 | Macro-F1 | Micro-F1 | Macro-F1 | Micro-F1 | Macro-F1 | Micro-F1 |
|---|---|---|---|---|---|---|---|---|---|
| base model few-shot | BERT.base | 2.40 | 3.06 | 2.08 | 3.18 | 9.75 | 16.31 | 7.26 | 8.39 |
| | BERT.large | 2.89 | 4.97 | 2.83 | 3.44 | 4.44 | 15.32 | 1.07 | 2.28 |
| | SciBERT | 4.70 | 10.01 | 5.14 | 6.51 | - | - | - | - |
| | LinkBERT | 0.81 | 1.32 | 1.45 | 1.77 | 10.45 | 36.06 | 0.16 | 10.90 |
| | Llama2-7B | 9.3 | 11.43 | 8.76 | 8.64 | 38.41 | 60.01 | 32.26 | 49.14 |
| $\epsilon = \infty$ | BERT.base | 2.02 | 2.88 | 1.88 | 2.23 | 29.05 | 31.37 | 17.50 | 19.81 |
| | BERT.large | 6.88 | 11.57 | 4.90 | 5.32 | 26.31 | 35.59 | 23.53 | 24.42 |
| | Llama2-7B | 14.97 | 18.77 | 11.52 | 10.85 | 32.94 | 50.65 | 57.53 | 63.15 |
| $\epsilon = 4$ (Ours) | BERT.base | 3.61 | 8.49 | 2.40 | 4.74 | 23.42 | 26.43 | 17.87 | 18.63 |
| | BERT.large | 6.31 | 11.16 | 3.07 | 6.45 | 16.77 | 22.98 | 21.71 | 22.67 |
| | Llama2-7B | 16.55 | 18.59 | 13.56 | 13.29 | 35.43 | 54.85 | 44.74 | 50.47 |

classification, which has been observed by Xie et al. (2023) in the non-private relational learning setting and by Li et al. (2021) in the private non-relational setting.

## 4.3 UTILITY, PRIVACY AND COMPUTATIONAL EFFICIENCY TRADE-OFFS

In this section, we study the trade-offs between utility, privacy, and complexity in private relational learning. We first investigate the hyperparameters of negative sampling, batch size, and learning rate in a realistic setting, where the training steps are fixed. Fig. 2 (Left) shows the impact of negative sampling in relational learning: increasing $k$ generally improves model performance while with a rapidly decreasing marginal benefit. To achieve a trade-off between performance and complexity, the optimal region is located at $k \in [4, 8]$. Fig. 2 (Middle) shows the effect of batch size $b$ on different models under the same privacy parameters: larger $b$ leads to better model performance and quick convergence, especially for Llama2-7B. This observation is consistent with non-relational private learning, where increasing $b$ achieves a better signal-to-noise ratio between the sum of clipped gradients and the Gaussian noise added in Eq. (3). The joint effect of batch size $b$ and learning rate $\eta$ is further studied and depicted in Fig. 3 (Left), Appx. E: larger batches and learning rates together lead to good performance under fixed training steps, which echoes the findings in privately fine-tuning LLMs on standard text data (Li et al., 2021). The main obstacle to using larger $b$ is the linearly increased computational and memory cost in privacy computing.

Next, we study how privacy parameters impact model utility. Fig. 2 (Right) plots the privacy-utility curve of BERT.base on zero-shot relation prediction over MAG-USA/CHN datasets under different privacy budgets $\epsilon$ by adjusting noise multiplier $\sigma$ while keeping other parameters constant. In this case, the scale of privacy noise solely determines the privacy leakage, where the model performance decays proportionally to the increased noise added to clipped gradients. The norm clipping threshold $C$ does not affect the privacy budget $\epsilon$ here, but is crucial to the utility performance of DP models (Bu et al., 2024), and its impact on relational learning tasks is shown in Fig. 3 (Right), Appx. E. Picking a threshold $C$ that is larger than the actual gradient norm means that most clipping in Eq. (3) is not effective, and the noise $\sigma C$ is added more than necessary. In general, small values of $C$ work better for relational learning, which aligns with the general practice and observation of DP learning on non-relational data in both vision and language tasks (Tramer & Boneh, 2020; Li et al., 2021).

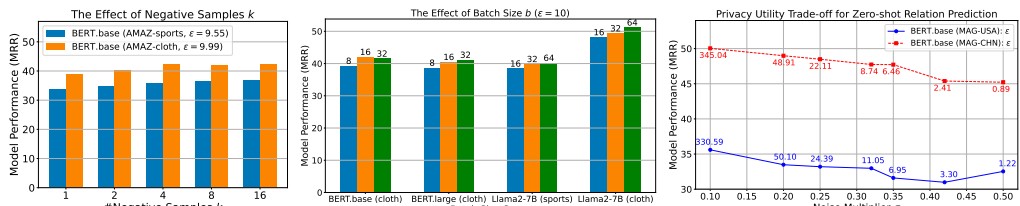

Figure 2: Effects of negative sample $k$, batch size $b$, and noise multiplier $\sigma$ in private relational learning for zero-shot relation prediction.

# 5 RELATED WORK

**LLMs with Relational Data** Extensive work has focused on using relational data to enhance foundation models, especially for fine-tuning LLMs on graphs. These methods can be classified into two types. *Objective-based:* Yasunaga et al. (2022b); Duan et al. (2023); Xie et al. (2023) proposed to associate entity representations from LLMs with relational information by optimizing the objective based on specific graph tasks. E.g., relation prediction is a typical task in unsupervised graph learning, as adopted in this work. *Graph-encoder-based:* Chien et al. (2021); Yasunaga et al. (2022a); Zhu et al. (2023); Xie et al. (2023); Jin et al. (2023b) pair LLMs with a graph encoder (e.g., GNNs (Kipf & Welling, 2017)) to incorporate relational information in an end-to-end manner, where LLMs act as feature extractors for textual attributes, and their output with associated relations is fed into GNNs for aggregation and prediction. These models may be privatized by combining the approach proposed in this work with the privatized method for GNNs (Sajadmanesh et al., 2023; Chien et al., 2024), though the entire pipeline could be complex and beyond the scope of this work.

**Privacy-preserving Graph Learning** Significant research has focused on privacy-preserving graph embedding and learning algorithms with DP guarantees (Li et al., 2023b). Daigavane et al. (2021) proposed a privacy-preserving approach for training GNNs via extensions of DP-SGD. Olatunji et al. (2021) adopted teacher-student models to enable the DP release of GNNs. Sajadmanesh et al. (2023) improved utility-privacy trade-offs by decoupling feature propagation and network training, and their work further got extended in subsequent studies (Sajadmanesh & Gatica-Perez, 2024; Chien et al., 2024). These methods specialize in generating private node representations, which do not mitigate privacy risks when relations that involve multiple entities are used for supervision.

**Privacy-preserving for LLMs** Data privacy in LLMs focuses on safeguarding sensitive information that could be exposed during operations (Yao et al., 2024). Recent efforts have utilized DP-SGD for both pretraining and fine-tuning LLMs. For instance, Anil et al. (2021) trained a privacy-preserving BERT-Large model from scratch. However, due to the resource-intensive nature of LLMs, the focus has shifted towards private fine-tuning of publicly pretrained models. Hoory et al. (2021) explored private full fine-tuning of BERT models with domain-specific data, while further advancements in this field include the works of Basu et al. (2021); Kerrigan et al. (2020); Senge et al. (2021); Li et al. (2021). There is also growing interest in efficient fine-tuning techniques. Yu et al. (2021) applied parameter-efficient fine-tuning (PEFT) methods for private fine-tuning of LLMs, and Li et al. (2021) introduced ghost clipping to accelerate gradient clipping in DP-SGD. However, these methods primarily address privacy concerns for standard text data. In contrast, our work extends these privacy-preserving approaches to relational data, filling an important gap in this research area.

# 6 CONCLUSION

Leveraging relational data to enhance AI models holds great promise. This work proposes a novel privacy-preserving training pipeline that addresses the unique privacy and computational challenges in relational learning by decoupling the dependencies in sampled relations for training and exploiting the structure of individual gradients for efficient clipping. We consider scenarios frequently encountered in applying relational learning to fine-tune pretrained models and enforce privacy guarantees on the relationships used for training. Our study on private relational learning shows that fine-tuning pretrained language models with our approach can significantly improve their performance on new test domains while keeping the relational data used for training private. We further explore the privacy, utility, and computational efficiency trade-offs and conduct an extensive study on hyperparameter selection for private learning on relational data.

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

## A  MORE RELATED WORK

**Private Graph Embedding Methods** Graph embedding encodes nodes into low-dimensional vectors, preserving topological information (Hamilton et al., 2017). Xu et al. (2018) proposed a private network embedding method using objective perturbation in DeepWalk (Perozzi et al., 2014) but faced scalability issues for complex sensitivity calculations. Zhang & Ni (2019) addressed these issues by applying a Lipschitz condition (Raskhodnikova & Smith, 2016) and gradient clipping. Epasto et al. (2022); Wei et al. (2024) studied DP PageRank methods, which can be leveraged to generate DP graph embedding as well. These methods specialize in preventing privacy leakage during generating node embeddings but do not mitigate privacy risks when relations that involve multiple entities are used for supervision.

**Contrastive Learning with Differential Privacy** Existing studies on private contrastive learning aim to eliminate the risk of sample correlation in contrastive loss and thus protect the privacy of individual training samples. Li et al. (2022) proposed to add privacy noise to the similarity matrix between pairs of inputs to reduce the sensitivity of gradients w.r.t. the contrastive loss. Kong et al. (2023) extended it to similarity-based loss functions by bounding the pairwise similarity gradients. Bao et al. (2024) proposed to train vision models with the mixup technique under DP by leveraging augmentation multiplicity. These methods focus on privately learning representations of non-relational samples by contrastive views but cannot be used to address the privacy challenge of relation coupling in training models on relational data.

## B  STANDARD DP LEARNING PIPELINES

DP-SGD (see Alg. 2) (Song et al., 2015; Abadi et al., 2016) is proposed for training deep learning models on (non-relational) samples with a privacy guarantee. DP-Adam (see Alg. 3) works similarly as regular Adam (Kingma & Ba, 2014) but performs updates and moment accumulation with privatized gradients. The gradient privatization part is the same as that performed in DP-SGD, where the privacy analysis and guarantees for DP-SGD still hold for DP-Adam due to the post-processing property of DP (Dwork et al., 2014).

## C  EXPERIMENTAL DETAILS

**Datasets** Item and paper titles are used as textual attributes associated with the entities in the Amazon e-commerce network (AMAZ) (McAuley et al., 2015)) and the Microsoft Academic Graph (MAG) [2] (Sinha et al., 2015), respectively. OpenAlex API [3] (Priem et al., 2022) is used to obtain metadata of papers in MAG as the Microsoft Academic service has been retired. For some items/papers, we concatenate their titles with the corresponding description/abstract following Jin et al. (2023b), since the title is too short. The max length of the input sequence $M$ is set to 32. The semantics of relational information used for supervision are "item-co-purchased-item" and "paper-cited-paper" for AMAZ and MAG networks, respectively. To mimic the case in cross-category recommendation, two subgraphs are selected from AMAZ to that only contain items belonging to

---

[2] ODC-BY License, refer to `https://opendatacommons.org/licenses/by/1-0/`
[3] CC0 License, refer to `https://creativecommons.org/public-domain/cc0/`

---

**Algorithm 2:** DP-SGD from Abadi et al. (2016)

---

**Input:** Training data $x_1, \ldots, x_N$, loss function $\mathcal{L}(\Theta) = \frac{1}{N} \sum_i \mathcal{L}(\Theta, x_i)$; Parameters: learning rate $\eta_t$, batch size $b$, gradient norm threshold $C$, noise multiplier $\sigma$ or privacy budget $\epsilon$.

**Initialize** find the optimal value of $\sigma$ via calibration if $\epsilon$ is given.

**for** $t = 1$ **to** $T$ **do**
    **Subsampling**
Randomly sample $\mathcal{B}_t$ with sampling probability $b/N$
    **Compute Gradient**
For each $x_i \in \mathcal{B}_t$, compute $\mathbf{g}_t(x_i) \leftarrow \nabla_{\Theta_t} \mathcal{L}(\Theta_t, x_i)$
    **Gradient Clipping**
$\bar{\mathbf{g}}_t(x_i) \leftarrow \mathbf{g}_t(x_i) / \left[ \max\left(1, \frac{||\mathbf{g}_t(x_i)||_2}{C}\right) \right]$
    **Add Noise**
$\tilde{\mathbf{g}}_t \leftarrow \frac{1}{b} \left[ \sum_i \bar{\mathbf{g}}_t(x_i) + \mathcal{N}(0, \sigma^2 C^2 \mathbf{I}) \right]$
    **Parameter Update**
$\Theta_{t+1} \leftarrow \Theta_t - \eta_t \tilde{\mathbf{g}}_t$
**end for**
**Output** $\Theta_T$ and calculate the overall privacy cost $(\epsilon, \delta)$ using an accounting method if $\sigma$ is given.

---

**Algorithm 3:** DP-Adam (Kingma & Ba, 2014; Abadi et al., 2016)

---

**Input:** Training data $x_1, \ldots, x_N$, loss function $\mathcal{L}(\Theta) = \frac{1}{N} \sum_i \mathcal{L}(\Theta, x_i)$; Parameters: learning rate $\eta_t$, batch size $b$, gradient norm threshold $C$, noise multiplier $\sigma$ or privacy budget $\epsilon$, initial moment estimates $m_0, v_0$, exponential decay rates $\beta_1, \beta_2$, avoid division-by-zero constant $\gamma$.

**Initialize** find the optimal value of $\sigma$ via calibration if $\epsilon$ is given.

**for** $t = 1$ **to** $T$ **do**
    **Subsampling**
Randomly sample $\mathcal{B}_t$ with sampling probability $b/N$
    **Compute Gradient**
For each $x_i \in \mathcal{B}_t$, compute $\mathbf{g}_t(x_i) \leftarrow \nabla_{\Theta_t} \mathcal{L}(\Theta_t, x_i)$
    **Gradient Clipping**
$\bar{\mathbf{g}}_t(x_i) \leftarrow \mathbf{g}_t(x_i) / \left[ \max\left(1, \frac{||\mathbf{g}_t(x_i)||_2}{C}\right) \right]$
    **Add Noise**
$\tilde{\mathbf{g}}_t \leftarrow \frac{1}{b} \left[ \sum_i \bar{\mathbf{g}}_t(x_i) + \mathcal{N}(0, \sigma^2 C^2 \mathbf{I}) \right]$
    **Parameter AdamUpdate**
$m_{t+1} \leftarrow \beta_1 \cdot m_t + (1 - \beta_1) \cdot \tilde{\mathbf{g}}_t,\ v_{t+1} \leftarrow \beta_2 \cdot v_t + (1 - \beta_2) \cdot \tilde{\mathbf{g}}_t^2$
$\hat{m}_{t+1} \leftarrow m_{t+1}/(1 - \beta_1^t), \hat{v}_{t+1} \leftarrow v_{t+1}/(1 - \beta_2^t)$
$\Theta_{t+1} \leftarrow \Theta_t - \eta_t \cdot \hat{m}_{t+1} / \left( \sqrt{\hat{v}_{t+1}} + \gamma \right)$
**end for**
**Output** $\Theta_T$ and calculate the overall privacy cost $(\epsilon, \delta)$ using an accounting method if $\sigma$ is given.

---

the category of clothing (AMAZ-Cloth) and sports (AMAZ-Sports). For entity classification, the class names of the AMAZ dataset are listed in Table 5. Based on the geographic metadata of paper-authors, we select two subgraphs from MAG containing papers written by authors from the United States (MAG-USA) and China (MAG-CHN) to simulate the case in cross-regional model deployment. The coarse-grained class of papers is refined by selecting Top-K-occurrence of 349-class obtained from Open Graph Benchmark [4] (Hu et al., 2020) and merging the other classes into one.

**Environment** We use a server with two AMD EPYC 7543 CPUs, 512GB DRAM, and NVIDIA Quadro RTX 6000 (24GB) GPUs for BERT-based models and A100 (80GB) GPUs for Llama2-7B models. The codebase is built on PyTorch 2.1.2, Transformers 4.23.0, PEFT 0.10.0, and Opacus 1.4.1. The source code is attached and should be paired with the Transformers and PEFT packages from HuggingFace and the Opacus library specified above.

---

[4]ODC-BY License

Table 5: Class names of the AMAZ dataset.

| AMAZ-Cloth | | AMAZ-Sports | |
|---|---|---|---|
| Label | Name | Label | Name |
| 0 | girls | 0 | accessories |
| 1 | men | 1 | action sports |
| 2 | novelty | 2 | boating & water sports |
| 3 | luggage | 3 | clothing |
| 4 | baby | 4 | cycling |
| 5 | fashion watches | 5 | baby |
| 6 | shoes | 6 | exercise & leisure sports |
| 7 | boys | 7 | fan shop |
| 8 | adidas | 8 | golf |
| | | 9 | hunting & fishing & game room |
| | | 10 | outdoor gear |
| | | 11 | fitness |
| | | 12 | paintball & airsoft |
| | | 13 | racquet sports |
| | | 14 | snow sports |
| | | 15 | team sports |

Table 6: Hyperparameter search range for different models.

| Target Model | BERT.base | BERT.large | Llama2-7B |
|---|---|---|---|
| DP Guarantee $(\epsilon, \delta)$ | $(\text{-},1/|\mathcal{E}_{\text{train}}|)$ | $(\text{-},1/|\mathcal{E}_{\text{train}}|)$ | $(\text{-},1/|\mathcal{E}_{\text{train}}|)$ |
| Clipping threshold $C$ | 1 | 1 | 1 |
| Noise multiplier $\sigma$ | [0.3, 0.5] | [0.3, 0.5] | [0.3, 0.5] |
| LoRA rank $r$ | {2,4,8,16} | {2,4,8,16} | {2,4,8,16} |
| LoRA alpha $\alpha$ | 16 | 16 | 16 |
| LoRA dropout | [0, 0.2] | [0, 0.2] | [0, 0.2] |
| Target module | query, key, value, dense | query, key, value, dense | q_proj, v_proj |
| Batch size $B$ | {8, 16, 32, 64} | {8, 16, 32, 64} | {12, 16, 32, 64, 128} |
| Learning rate $\eta$ | $[10^{-4}, 10^{-6}]$ | $[10^{-4}, 10^{-6}]$ | $[10^{-4}, 10^{-6}]$ |
| LR scheduler | linear | linear | cosine |
| Weight decay $\lambda$ | $[0, 10^{-3}]$ | $[0, 10^{-3}]$ | 0 |
| Negative sample $k$ | {4, 6, 8, 16} | {4, 6, 8, 16} | {4, 8, 12, 16} |

Table 7: Model card of pretrained language models.

| Backbone Model | License | Model Card |
|---|---|---|
| BERT.base | Apache License 2.0 | https://huggingface.co/google-bert/bert-base-uncased |
| BERT.large | Apache License 2.0 | https://huggingface.co/google-bert/bert-large-uncased |
| SciBERT | Apache License 2.0 | https://huggingface.co/allenai/scibert_scivocab_uncased |
| LinkBERT.large | Apache License 2.0 | https://huggingface.co/michiyasunaga/LinkBERT-large |
| Llama2-7B | Meta Community License | https://huggingface.co/meta-llama/Llama-2-7b-hf |

Table 8: Privacy loss $\epsilon$ of model fine-tuning (LoRA rank $r$) on relational data.

| Privacy | Model | MAG-USA | | | MAG-CHN | | | AMAZ-Cloth | | | AMAZ-Sports | | |
|---|---|---|---|---|---|---|---|---|---|---|---|---|---|
| | | $\sigma$ | $\epsilon$ | $r$ | $\sigma$ | $\epsilon$ | $r$ | $\sigma$ | $\epsilon$ | $r$ | $\sigma$ | $\epsilon$ | $r$ |
| | BERT.base | 0.32 | 9.95 | 4 | 0.32 | 8.74 | 2 | 0.3 | 9.71 | 2 | 0.3 | 9.06 | 8 |
| $\epsilon = 10$ | BERT.large | 0.34 | 8.72 | 4 | 0.33 | 8.56 | 2 | 0.3 | 9.94 | 8 | 0.32 | 7.69 | 8 |
| | Llama2-7B | 0.378 | 7.91 | 4 | 0.357 | 8.16 | 4 | 0.326 | 8.50 | 8 | 0.315 | 8.83 | 8 |
| | BERT.base | 0.42 | 3.30 | 8 | 0.4 | 3.99 | 2 | 0.4 | 3.34 | 2 | 0.4 | 2.65 | 2 |
| $\epsilon = 4$ | BERT.large | 0.42 | 3.82 | 4 | 0.41 | 3.32 | 2 | 0.376 | 4.00 | 8 | 0.4 | 3.27 | 2 |
| | Llama2-7B | 0.456 | 3.97 | 4 | 0.433 | 4.00 | 4 | 0.4 | 4.00 | 8 | 0.4 | 3.88 | 8 |

**Private Fine-tuning** We use DP-Adam, a variant from DP-SGD, as the default optimizer for updating model parameters in a privacy-preserving manner: given privacy parameters of noise multiplier $\sigma$ (or calibrated $\sigma$ if $\epsilon$ is explicitly provided) and gradient norm clipping threshold $C$, with a learning rate $\eta$ from `1e-4` to `1e-6`, first 10% as warm-up steps, weight decay from `0` to `1e-3`. The test batch is set to 256 for relation prediction, which follows Jin et al. (2023b) that uses in-batch negatives for computing ranking metrics. For large pretrained models where the desired batch size exceeds the physical memory limit of GPU VRAM, we use gradient accumulation over multiple mini-batches to simulate training at the expected batch size. We search optimal training hyperparameters with the InfoNCE loss under given privacy parameters, where their ranges are summarized in Table 6. All pretrained model weights are publicly available and directly downloaded from Huggingface under proper licenses listed in Table 7.

**Inference Setting** Once the model is privately fine-tuned on relational data, it is deployed for inference under two settings for relation prediction and entity classification on the corresponding test domains (see Table 1):

- *Zero-shot*, where the model is directly used without further training on samples from the test domain. This setting is only applied for relation prediction: the dot product between entity embeddings is used as the scoring function for inference, which contains no additional parameters.
- *Few-shot*, where limited labels from the test domain are provided to further fine-tune the target models obtained after private relational learning. This setting is used for both relation prediction and entity classification (the classifier requires some labels for initializing parameters), corresponding to the data scarcity scenario from the test domain and limited resources to perform full domain-specific fine-tuning.

# D DETAILS FOR STUDIES IN SECTION 4.2

After privately fine-tuning target models on realtions from the training graph, we use the PRV accounting (Gopi et al., 2021) to track privacy loss and convert it to $(\epsilon, \delta)$-DP. Table 8 summarizes the values of noise multiplier $\sigma$ used and the actual privacy loss $\epsilon$ on relational data used for training each model one epoch, which corresponds to the results reported in Table 2 under the zero-shot setting and in Tables 3, 4 under the few-shot setting. Models under the few-shot setting have the same privacy loss as zero-shot since the examples used for further fine-tuning are non-private from the test domain. The scale of noise $C\sigma$ determines the privacy budget in DP-SGD, where higher privacy noise leads to lower privacy leakage $\epsilon$. Training with the same scale of privacy noise may result in different $\epsilon$ reported in Table 8: different batch sizes $b$ (sampling ratio $p = b/|\mathcal{E}|$) and numbers of iterations $T$ used in training affect the privacy accounting in DP-SGD (Balle & Wang, 2018).

# E ADDITIONAL RESULTS FOR STUDIES IN SECTION 4.3

Fig. 3 (Left) shows the joint effect of learning rate $\eta$ and batch size $b$ for BERT.base over zero-shot relation prediction on AMAZ-cloth under the same privacy parameters: larger batches and learning rates together lead to good performance (diagonal area) under fixed training steps. This observation aligns with the findings in privately fine-tuning LLMs on standard text data (Li et al., 2021). Fig. 3 (Right) shows the impact of norm clipping threshold $C$ for BERT.base on zero-shot

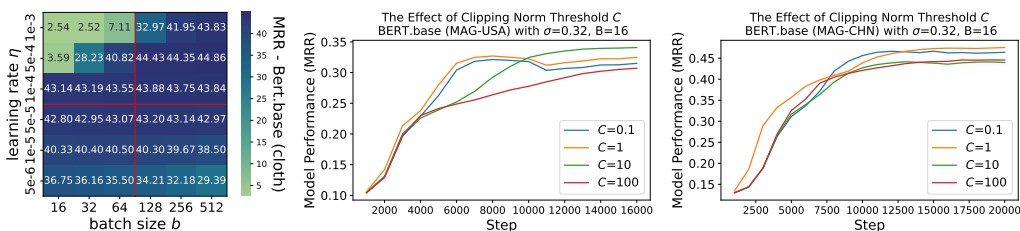

Figure 3: Effects of learning rate $\eta$ and batch size $b$ (Left), and clipping norm threshold $C$ (Right) in private relational learning for zero-shot relation prediction.

relation prediction over MAG-USA/CHN datasets, while other hyperparameters remain the same. The threshold $C$ does not affect the privacy budget $\epsilon$ here but is crucial to the utility performance of DP models (Bu et al., 2024). Picking a threshold $C$ larger than the actual gradient norm means that most clipping in Eq. (3) is not effective, and the scale of noise $\sigma C$ is added more than necessary. E.g., $C = 100$ always performs the worst in Fig. 3 (Right). In general, small values of $C$ work better for relational learning as suggested in the general practice and observation of DP learning on non-relational data (Tramer & Boneh, 2020; Li et al., 2021).

# F   SCOPE AND LIMITATION

Our proposed pipeline aims at private relational learning, with applications in fine-tuning pretrained large models when relational information indicated by graph edges is used for supervision in a privacy-preserving manner. Due to resource constraints and the intensity of privacy computing for large pretrained models, we choose parameter-efficient fine-tuning over full parameter fine-tuning. Our privacy setting only targets protecting the relationships between entities used for training and assumes the pretrained model weights are risk-free since they are publicly accessible. The two relational datasets of e-commerce and academic networks used in the experiment are open source and widely adopted in the community. The textual attributes associated with them are the titles and descriptions of shopping items and the titles and abstracts of research papers, respectively, and neither of them contains harmful, offensive, or biased language.

