# OpenReview forum: "Privately Learning from Graphs with Applications in Fine-tuning Large Pretrained Models"
_ICLR.cc/2025/Conference — ICLR 2025 Conference Withdrawn Submission_

### Official Review · Reviewer_DTgz · 2024-10-24

**Soundness:** 2
**Presentation:** 3
**Contribution:** 2
**Rating:** 5
**Confidence:** 4

**Summary:**

This paper investigates the potential privacy leakage issue in relational learning, where DP-SGD is not applicable since it violates the per-sample decoupling assumption. Rather than random or in-batch negative sampling, the authors propose to use a simple negative sampling strategy: pairing one end of the positive relation with entities uniformly sampled from the whole entity set, which is compatible with DP-SGD. The authors apply this method to privately fine-tune LLMs using real-world relational data as proof of concept.

**Strengths:**

1. The privacy issue in relational learning is an important research topic with a very broad impact.
2. This paper is generally well written and clearly organized.

**Weaknesses:**

1. The novelty of this paper is quite limited. The simple negative sampling solution proposed by the authors has already been widely used by many relational learning methods. This paper only contributes by highlighting its advantage in preserving privacy.
2. The technical quality of this paper is not good enough. The important privacy attack experiments are missing.

**Questions:**

1. Many relational learning methods already use the simple negative sampling strategy described in this paper. Most of them choose it because of its simplicity and efficiency. This paper merely highlights its advantage in preserving privacy when combined with DP-SGD. This seems to be the only innovation of this paper, which is inadequate.
2. The paper is missing some very important privacy attack experiments. The authors are advised to conduct membership inference attacks on different baselines (random/in-batch/proposed negative sampling with/without DP-SGD) and report the attack results. This can help validate whether the proposed negative sampling method indeed help DP-SGD in protecting relation privacy.
3. It is quite difficult to understand the difficulty of computing per-sample gradients for relational learning. Why must we first compute the gradients for individual entities (or even individual tokens) and then calculate the per-tuple gradients based on them? Why can't we just directly calculate the per-tuple gradients (which seems very simple in terms of programming)? It would be better if the authors could elaborate on this with more detailed explanations or figure illustrations.
4. In Table 4, why can the privately fine-tuned version sometimes outperform the non-privately fine-tuned baseline?
5. There are a few typos, such as
    1. Line 167: $e^{-}i_j$ should be corrected as $e^{-}_{i_j}$
    2. Line175-176: the Hinge loss formulation seems incorrect, it should be $[\gamma+\mathbf{z}_{e^{-}_{i_j}}-\mathbf{z}_{e_i^+}]_+$
    3. Line 292: "where $d$, $q$ are" should be corrected as "where $d$, $p$ are"

---

### Official Review · Reviewer_TTSu · 2024-11-05

**Soundness:** 2
**Presentation:** 1
**Contribution:** 2
**Rating:** 3
**Confidence:** 3

**Summary:**

This paper works on improving the privacy on relational learning. Experiments on applying LLM for relational learning with lora and differential privacy show the performance.

**Strengths:**

1. Relational learning is an important topic.
2. DP-SGD can improve the privacy preservation during training.
3. Experiments show LLM can have good performance in relational learning.

**Weaknesses:**

1. Beside using DP-SGD+Lora for relational learning, the technical contribution is unclear.
2. Experiments did not compare with former GNN based methods.
3. The writing can be improved.

**Questions:**

As in weaknesses, the main concern is the technical contribution. What is the improvement of the proposed method over DP-SGD?

---

### Official Review · Reviewer_ZLDE · 2024-11-09

**Soundness:** 2
**Presentation:** 3
**Contribution:** 2
**Rating:** 5
**Confidence:** 3

**Summary:**

This paper focuses on relational learning for graphs, tackling the challenge in existing privacy-preserving methods, where approaches (like DP-SGD) rely on a gradient decoupling assumption that doesn’t hold in graphs. To address it, this paper proposes a relational learning pipeline that can decouple dependencies in sampled relations, making it possible to adapt these privacy-preserving methods. Specifically, it introduces a decoupled sampling strategy for relations, ensuring that perturbation on a relation affects at most one loss term at a time. This approach is applied to fine-tuning LLMs on sensitive graph data.

**Strengths:**

S1. This work claims to be the first to explore privacy-preserving methods for relational learning, filling a gap where previous works focus only on entity-level privacy in graphs.

S2. Besides proposing a decoupled sampling strategy to preserve the privacy of relations, it also tackles the efficiency challenge in fine-tuning LLMs in the private relational learning setting, by leveraging low-rank characterization of per-sample gradients.

S3. The authors discuss trade-offs among utility, privacy, and computational efficiency in private relational learning.

S4. This paper is well-organized and easy to follow.

**Weaknesses:**

W1. While the paper addresses a general challenge—decoupling positive and negative relation sampling to enable privacy-preserving methods (i.e., DP-SGD)—the setting seems to be specific and trivial, involving aspects like LLMs, domain transfer, and cold-start (missing relations). It doesn’t clearly justify how the proposed sampling method would apply to the basic setting without more assumptions. For instance, it’s important to study whether the decoupled sampling method effectively supports privacy-preserving relational learning on the perspective of graph itself.

W2. The authors state in the introduction that current GNN methods fail to mitigate privacy risks in relational learning, yet it’s unclear how this extends to relational learning with LLMs. The reason "since many relational datasets involve entities with rich textual attributes" is not convincing. For example, with rich text, we could also encode it with LLMs and train graph ML models. Why is it necessary or what is the benefits to use LLMs to learn relations and to study privacy preserving specifically in this context?

W3. Regarding the random negative sampling mentioned in the last paragraph of Section 3.1, besides edge-based random sampling, another approach is to sample end nodes, which aligns with the approach discussed in the first paragraph of Section 3.2. This idea is thus not novel.

W4. (1) In negative sampling, it’s crucial to exclude positive edges. The authors state, “Note that this pairing strategy may generate negative relations (u, v) that are actually positive relations (u, v) ∈ E but with a low probability. Fortunately, our experiments show this does not obviously hurt the model performance.” However, this observation is based on empirical experiments limited to the specific setting. Also, whether the probability is low depends on the number of relations. Therefore, it is less rigorous to make this claim. Additionally, the impact of sampling positive edges as negatives can vary across cases.
(2) If positive relations are sampled as negatives, although “removing or adding a positive relation will change at most one tuple  $E_i$  in a mini-batch,” it could affect multiple message-passing steps or lead to conflict training signals, meaning that the sensitivity might not be bounded.
(3) On the other hand, if positive edges are excluded from the negative sampling, it cannot ensure that the perturbation of that positive relation impact at most one tuple, which still presents the challenge mentioned in Section 3.1. This point needs discussion in the paper.

W5. In Section 3.3, the authors mention limitations of two prior works, but there are more recent studies on efficiency with DP-SGD that should be reviewed and discussed to see if they apply to this relational learning context.

W6. Since entity features are usually also sensitive, this work is limited by considering only the privacy of relations, raising concerns about the privacy of features.

W7. The datasets (MAG and AMAZ) are small- and medium-scale. It would be better to include experiments on large-scale graphs.

W8. From the result tables, there are obvious performance drops comparing to the non-private preserving baselines.

W9. It would be better to also include empirical efficiency analysis.

**Questions:**

Q1. How would the decoupling sampling strategy apply to traditional graph ML methods?

Q2. This work aims to simulate common scenarios in relational learning, so why does it focus on domain transfer and missing relations in the new domain? How will it work when there exists domain shifts, and how well can relational learning transfer across domains? It would also be interesting to see performance evaluated on the same domain.

Q3. In real scenarios, the data is usually represented by heterogeneous graphs with multiple entity and relation types. How would the proposed pipeline work for such scenarios?

---

### Note · Authors · 2024-12-17

**Comment:**

We deeply appreciate the time and feedback of all reviewers. There is consensus that our work addresses an important and underexplored research topic: privacy-preserving relational learning on graph data. We pioneered a general pipeline to extend DP-SGD to relational learning tasks and tackle challenges in privacy computation for large models.

Reviewers ZLDE and DTgz both acknowledged our efforts in addresssing a key challenge and noted the broad potential impact of our work. Reviewer ZLDE questioned the generality of our experimental setup and the applicability to traditional graph ML methods. Reviewer DTgz raised concerns about privacy attack experiments and per-sample gradient computation. Reviewer TTSu questioned the technical contributions of the paper. We felt it necessary to clarify some misunderstandings and delve deeper into some of the key concerns.

**Technical Contribution Clarification**
This work bridges relational learning and privacy research. We identified the limitation that prevents the application of DP-SGD to relational learning: coupled sampling of positive and negative relations, violating the per-sample decoupling assumption of DP-SGD. The key finding is to decouple the sampling, where a simple yet effective way is constructing negative relations from uniformly sampled nodes without access to any relational information. This enables per-tuple gradient clipping, ensuring theoretical compatibility with DP-SGD. Combined with efficient gradient control for large models, this pipeline extends DP-SGD to learn sensitive relations from graphs. We also provide extensive experiments to evaluate the trade-offs between privacy, utility, and computational efficiency in relational learning tasks.

**Experimental Settings**
Our privacy-preserving training pipeline is general and not tied to a specific model, architecture, or scenario. We choose LLMs as a proof of concept to showcase the effectiveness of relational learning under privacy preservation. LLMs provide inductive inference and generalization capabilities in zero-/few-shot settings, where traditional GNNs often struggle. However, when the accessed topology is protected (e.g., DP-GNN), our approach is compatible with graph ML models for relational learning tasks formulated as in Eq. (1), including applications on heterogeneous and knowledge graphs.

**Experiments on Privacy Attacks**
Differential privacy provides provable guarantees against adversarial inference attacks, and several works have used membership attacks to empirically audit DP bounds on the success of attacks, especially those obtained from DP-SGD. While previous studies on DP fine-tuning of LLMs (Li et al., 2021; Yu et al., 2021) did not conduct such experiments, we agree that performing empirical privacy attacks (e.g., MIA) on relational data would further strengthen our work.

**Gradient Control in Private Relational Learning**
Per-sample gradient computation relies on automatic differentiation libraries, but relational learning presents unique challenges. Each relation tuple involves multiple entities, with each entity consisting of multiple tokens. Current autograd algorithms only keep batch-averaged gradients, causing the main bottleneck. Existing private learning libraries utilize intermediate results of autograd algorithms hooked during the forward and backward passes to recover per-sample gradients batch-wisely. However, for sequence models, these algorithms track results at the token level, making per-tuple gradient computation inefficient. Our solution directly leverages token-level intermediate results to efficiently compute per-tuple gradients (lines 297-301), substantially reducing memory overhead.

We remain committed to advancing privacy-preserving relational learning and thank the reviewers for their feedback. Unfortunately, we have to withdraw this version to clarify and address the issues raised, incorporate valuable suggestions, and better convey our contribution.

**Withdrawal Confirmation:**

I have read and agree with the venue's withdrawal policy on behalf of myself and my co-authors.